# Influence of Abrasive Waterjet Parameters on the Cutting and Drilling of CFRP/UNS A97075 and UNS A97075/CFRP Stacks

**DOI:** 10.3390/ma12010107

**Published:** 2018-12-30

**Authors:** Raul Ruiz-Garcia, Pedro F. Mayuet Ares, Juan Manuel Vazquez-Martinez, Jorge Salguero Gómez

**Affiliations:** Department of Mechanical Engineering & Industrial Design, Faculty of Engineering, University of Cadiz, Av. Universidad de Cadiz 10, E-11519 Puerto Real-Cadiz, Spain; raul.ruizgarcia@uca.es (R.R.-G.); juanmanuel.vazquez@uca.es (J.M.V.-M.); jorge.salguero@uca.es (J.S.G.)

**Keywords:** AWJM, stack, CFRP, aluminum UNS A97050, SOM/SEM, kerf taper, surface quality, macrogeometric deviations

## Abstract

The incorporation of plastic matrix composite materials into structural elements of the aeronautical industry requires contour machining and drilling processes along with metallic materials prior to final assembly operations. These operations are usually performed using conventional techniques, but they present problems derived from the nature of each material that avoid implementing One Shot Drilling strategies that work separately. In this work, the study focuses on the evaluation of the feasibility of Abrasive Waterjet Machining (AWJM) as a substitute for conventional drilling for stacks formed of Carbon Fiber Reinforced Plastic (CFRP) and aluminum alloy UNS A97050 through the study of the influence of abrasive mass flow rate, traverse feed rate and water pressure in straight cuts and drills. For the evaluation of the straight cuts, Stereoscopic Optical Microscopy (SOM) and Scanning Electron Microscopy (SEM) techniques were used. In addition, the kerf taper through the proposal of a new method and the surface quality in different cutting regions were evaluated. For the study of holes, the macrogeometric deviations of roundness, cylindricity and straightness were evaluated. Thus, this experimental procedure reveals the conditions that minimize deviations, defects, and damage in straight cuts and holes obtained by AWJM.

## 1. Introduction

Over the last few decades, the aeronautical industry has been highlighted for its capacity to develop and manufacture structural elements built with advanced materials, having achieved a leading position in this area of activity with respect to other sectors.

In this sense, the aeronautical industry has demonstrated its capacity for the development and manufacture of complex elements built with advanced materials. Thus, the main manufacturers (Airbus and Boeing) have increased the use of new materials, mainly plastic matrix composites, in combination with those traditionally used, such as Duralumin alloys of 2XXX or the Al-Zn of 7XXX series, with the aim of reducing aircraft weight, maintaining the structural integrity of the assembly. These materials have undoubted advantages linked to the demand of greater safety, and lower energy consumption and maintenance costs that characterize the air-transport today. Additionally, they provide an excellent relationship between mechanical strength and weight, rigidity and an increase in the life-cycle thanks their good behavior against fatigue and corrosion [1,2].

Most of the structural elements used in aircraft construction need to undergo different machining operations, mainly drilling or milling of contours, prior to assembly work through rivets in the Final Assembly Lines (FAL) [3,4]. During the assembly tasks in aeronautical structures, these materials are joined in the form of stacks, which must be processed with drilling cycles under strict dimensional and geometric requirements, making it difficult to keep these tolerances under control when the nature of the materials is different [5,6,7,8].

Indeed, the combination of materials of a different nature has a negative impact during machining operations. On the one hand, both the heterogeneity of the material and the abrasive behavior of the carbon fiber negatively affect the tool life. Therefore, machining conditions and tool geometry must be adapted to these materials in order to reduce tool wear and thermal and mechanical defects produced during the cutting process, such as delamination or thermal damage to the composite matrix [9,10,11]. Moreover, Sorrentino et al. [12] demonstrated that Abrasive Waterjet Machining (AWJM) extends the high cycle fatigue strength of bolt holes and the fatigue life of bolted composite joints. On the other hand, aluminum alloys tend to modify the geometry of the tool [13], especially by the development of adhesive phenomena such as Build Up Layer (BUL) or Build Up Edge (BUE) [5,14]. The union of these phenomena causes accelerated wear of the tool through the loss of geometry and the increase in temperature reached during the cutting process, which causes a reduction in tool life due to the synergy of the wear mechanisms produced.

This is compounded by problems at the stack interface, such as burring and cleaning due to accumulated chip residues. As a result, the drilling process is complex to carry out in a single step [15]. Instead, different successive drilling steps must be carried out until the final diameter is obtained, including cleaning the rework at the interface, which does not allow One Way Assembly (OWA) to be achieved as a key technology for process automation.

Alternatively, some authors have conducted studies of machining stacks with unconventional technologies such as laser or AWJM [10,16,17,18,19]. In particular, AWJM has been widely studied as one of these machining alternatives to replace contour milling processes, although experimental studies are also beginning to appear, analyzing the influence of drilling on different aeronautical materials, Table 1. This is mainly due to different factors that positively affect the surface integrity of the final parts. Among them, and in comparison with conventional machining processes, the absence of tool wear, the reduction of residual stresses induced on the surface of the material and the reduction of surface thermal damage as a result of low cutting temperatures should be highlighted [10,20,21].

However, the AWJM process shows its own limitations that lead to the appearance of specific defects during the cutting process (Figure 1). The most common defects in the process are the kerf taper, the Erosion Affected Zone (EAZ) and the formation of three possible different roughness zones along the machined surface [22]:
-Initial Damage Region (IDR). The area where the water jet hits on the material producing EAZ. The roughness in this region is high due to the abrasive particles impacting the material.-Smooth Cutting Region (SCR). The region of variable thickness depending on the cutting parameters. It is the region with the best surface quality because it does not suffer the impact of particles and the jet still has enough kinetic energy to cut.-Rough Cutting Region (RCR). The final region where the jet ends of cut material. The jet has lost enough cutting capacity and produces macrogeometrical defects as striation marks.

Specifically, the removal of material through AWJM is produced by erosion caused by particles that impact the material at high velocity and affect each material differently. In the case of carbon fiber reinforced with plastic matrix, the formation of the erosion process produces the breakage of the fibers and the degradation of the matrix. This prevents the layers of the material from remaining bonded causing the formation of initial cracks that result in delaminations when abrasive particles penetrate between the layers of the composite [30].

However, some characteristic defects in the final part may occur as a result of the effect of the combination of different parameters. In this article a study based on the influence of the main cutting parameters on AWJM is carried out in order to reduce the appearance of the defects mentioned in stacks formed by the aluminum alloy UNS A97050 and Carbon Fiber Reinforced Plastic (CFRP). To this end, two experiments were carried out based on the operations most required in the machining of aeronautical structures: Straight cuts to analyze the cutting profile and drills to study the viability of the process. Finally, the state of the cuts was evaluated through the use of microscopic inspection techniques and macro and microgeometric deviations.

## 2. Materials and Methods

For the experimental development a CFRP AIMS 05-01-002 composite plate, Table 2, and a UNS A97075 aluminum alloy plate with a tensile strength of 496 MPa and a shear strength of 290 MPa have been used. Both 5 mm thickness plates have been mechanically joined by eight bolts to obtain two stack configurations: CFRP/UNS A97075 and UNS A97075/CFRP.

As technological parameters, combinations were made for each configuration of the three most significant parameters: Water pressure (WP), abrasive mass flow rate (AMFR) and traverse feed rating (TFR), due to the influence analyzed in [31]. The separation distance was kept constant at 3 mm throughout the experimental phase and the abrasive selected was garnet with an average particle size of 80 µm in order to optimize aluminum penetration [32]. Under these considerations, the experimental design based on levels shown in Table 3 was established.

To carry out the tests, two experimental blocks for each stack were made. On the one hand, straight cuts were made in order to study the influence on the kerf taper and the different roughness zones. On the other hand, 8 mm holes were drilled to study macrogeometry due to the fact that 7.92 mm is a common drill diameter used in the aeronautical industry. For this purpose, the experimental design and pre-simulation were carried out using the CAD/CAM software Lantek^®^ edition 34.02.02.02.02.02.02, making a total of 48 tests mechanized with a TCI water jet cutting machine model BPC 3020.

For the evaluation of straight cuts, on the one hand, optical evaluation of the machined material has been used by means of Stereoscopic Optical Microscopy (SOM) and Scanning Electronic Microscopy (SEM) techniques, and on the other hand, electron dispersive spectroscopy (EDS) was used to analyze the compositional state of the samples. A Nikon SMZ 800 stereo optical microscope was used for the SOM inspection and the Hitachi SU 1510 microscope was used for the SEM inspection. These techniques were used to study the incrustation of abrasive particles in the IDR zone and in the delaminations produced. In addition, it was used to generate a deeper measurement of the kerf taper. The literature tends to evaluate the taper as the difference between the cutting width of the water inlet and the cutting width of the water outlet depending on the thickness of the plate [19,20,33] as shown in Figure 2b. However, this process concurs in a high variability depending on two width measures (W_top_ and W_bottom_). Since the IDR may interfere with that extent, this paper proposes a new methodology based on image processing methods, for which ImageJ and Microsoft Excel^®^ software were used. It consists of capturing the image of the cut and its subsequent digitalization in 10 points with a non-linear distribution, as shown in Figure 2c. Then, a coefficient between W_top_ and W_bottom_ is usually obtained, however as it can be observed in Figure 2c, the representation of the cut would be unreal. Regarding Figure 2c, once you remove the IDR the shape of the cut is almost a vertical line. That is why, in this paper, measures to calculate the average width of the cut have been used.

This new methodology does not provide a coefficient as taper measure, but an average distance. A distance that represents, in a more realistic way, the profile of the cut. However, in order to obtain an accurate result, data from the IDR has to be rejected, as is already regarded by some authors [34]. That way, the cut depth was divided into 10 measures with cosinoidal distribution, ensuring more measures density near the top and the bottom. Then, the measures that maintained a height variation regarding the next measure have been disposed, that is the case of the upper measure and the second one in Figure 2c. Taking all other measures, an average is calculated and that is the kerf taper vale proposed.

For the evaluation of the holes, a station of measurement Mahr MMQ44 Form Tester (Mahr, Göttingen, Germany) was used to measure the roundness at the entrance and exit of the drill in each material, the cylindricity of the entire profile of the drill, and the straightness in four separate generatrices to 90° as seen in Figure 3a. To analyze the macrogeometric deviations, replicas of the holes due to the impossibility of direct measuring on the material were fabricated. These replicas were made with a polymer type F80 Ra (R.G.X, Plastiform, Madrid, Spain) with the ability to guarantee stability during the measurement process for diameters greater than 4 mm. It is a two component polymer that solidifies after mixture. Plastiform provides a tool that ensures correct mixture while the polymers are injected into the hole that was replicated. It is a manual process that leads to hole replicas after 10 minutes of polymer solidification.

For the measurement of roughness, the Mahr Perthometer Concept PGK 120 (Mahr, Göttingen, Germany) was employed, as shown in Figure 3. This measurement was focused on the parameter Average Roughness (Ra), since it is one of the most used roughness parameters in the literature.

Ra analysis performed to the specimens in each test was carried out in three different zones coinciding with IDR, SCR and RCR (Figure 4). That way, six measures were obtained for each test performed, making a total of 144 roughness measurements.

Finally, to distinguish the most significant parameters for evaluation results, analysis of variance (ANOVA) for a 95% confidence interval was employed. After that, contour charts for each variable studied in the experimental were obtained.

## 3. Results and Discussion

### 3.1. Straight Cuts Evaluation

#### 3.1.1. SOM/SEM Evaluation

SOM inspection was carried out of both jet entrances into the stack (Figure 5), and along the cut profile. This way, the jet variations contribution to the kerf profile can be observed, phenomenon related to damages produced in the IDR zone [31].

On the other hand, Figure 6 shows the profile of CFRP specimens in order to identify delaminations. In order to visualize the delamination along the machined surface, several images were taken showing the absence of visible delamination after machining in the test performed with the parameters considered to be the most aggressive.

Figure 7 shows the results of the SEM inspection in CFRP showing that no delamination was detected. However, Figure 7c shows in detail the state of the specimen entrance zone where signs of impact deformation and particle drag were observed. This state extends to the interface reflecting that a percentage of particles have lodged in the space between the two materials.

As for aluminum alloy, SOM study showed a series of dark colored streaks along the profile that repeated for both configurations to a greater or lesser extent depending on the energy of the jet. Specifically, Figure 8a shows the marks mentioned at the bottom while Figure 8b at the top. This phenomenon, together with the color of the stretch mark, seems to indicate that they are located in the zone close to the contact with the carbon fiber. Finally, Figure 8c shows the result of the study for test 11 where no transfer of carbon fiber to aluminum is observed, possibly due to the lower WP and AMFR and thus, lesser jet kinetic energy resulting in an inferior material removal rate [10].

In an attempt to obtain more information on the marks observed in Figure 8, the SEM/EDS inspection of aluminum was focused on discovering the state of the aluminum and the nature of these marks. Initially, Figure 9a,b shows the state of the material at the inlet. In a detailed way, the embedded particles and the deformation produced during the cutting process are appreciated, coinciding with the IDR or zone 1.

On the other hand, Figure 9c,d shows the stain examined in the striations observed by SOM microscopy and the results of the EDS analysis, respectively. The EDS analysis revealed the high presence of carbon at this point, confirming the carry-over of carbon particles during machining from one material to another. It should be noted that no traces of aluminum deposited on the carbon fiber were detected.

To analyze the state of the aluminum outside the zone of the stretch marks, another EDS spot was carried out outside those stains and showed almost no carbon and a huge peak on aluminum. As a direct conclusion, it appears that particles from composite are swept for the water beam and because of the water high energy, they end up embedded into UNS A97075. It seems like composite deposition over aluminum has a direct correlation with beam penetration capacity.

Therefore, contrary to what one would expect, a higher abrasive pressure and flow has not resulted in an increase in delaminations for both configurations. Similarly, the inclusion of abrasive particles has not greatly increased within the parameters studied. However, an increase in the inclusion of carbon particles in the aluminum alloy was observed as the pressure increases.

#### 3.1.2. Kerf Taper Evaluation

The ANOVA analysis performed showed that AMFR and TFR parameters were the most influential in taper formation. Average kerf taper values for each material when the configuration UNS A97075/CFRP is set are shown in Figure 10. The same values for configuration CFRP/UNS A97075 are shown in Figure 11. For deeper research, Appendix A shows the average taper for each test and its standard deviation.

Figure 10 shows a wide parameters combination that maintain kerf taper below 1.0 mm. This means a wide parameters combination that minimizes material removal percentage and leaves a more precise cut.

In this way, the data represented in Figure 10 and Figure 11 show that the taper is reduced as AMFR decreases and TFR increases, showing the best results for TFR = 45 mm/min and AMFR = 170 gr/min, in accordance with [33].

This behavior is shared with the CFRP behavior in CFRP/UNS A97075 configuration. Figure 11b, however, shows a very different behavior. This change is due to the lesser energy of the water beam when it collides with the aluminum. Since a percentage of energy is transformed during the CFRP machining, it appears that the AMFR is the determinant parameter when the cut’s width is examined. As for the differences between the two material configurations, Figure 11 shows that when the jet directly affects the carbon fiber, the taper generated for the best parameter ratio reaches values higher than 1.2 compared to the value 1 reached for the UNS A97075/CFRP configuration. This shows the difference in the mechanical properties of each material, offering greater resistance to penetration of the metallic material.

Overall, a similar behavior is observed between the materials located in the upper and lower part of the stack. Despite this, a smaller taper is always observed in UNS A97075 than in CFRP.

#### 3.1.3. Surface Roughness

The influential parameters in the analysis of surface quality are also AMFR and TFR for both configurations.

Figure 12 shows the results of the UNS A97075/CFRP configuration. A tendency to increase the roughness can be observed as TFR increases and AMFR decreases. Figure 13, on the other hand, shows the results of CFRP/UNS A97075 configuration. The same trend as in Figure 12 is observed although exist difference between the material placed at the top and bottom.

The data show that the AMFR parameter has a greater influence on aluminum, especially when it is at the exit of the material. This effect can be seen in the horizontality of the contour graph studied (Figure 13b). As for the composite material, it presents influence of TFR and AMFR for both configurations. In this sense, the data show that the composite material has slightly lower values than the metal alloy because the use of low pressures favors a better surface quality in CFRP to oppose less resistance to cutting. On the other hand, this means that the aluminum registers higher roughness data due to the low kinetic energy of the jet, favoring the appearance of defects in the different areas studied.

A more in-depth analysis of the data based on Appendix B reflects that the area with the greatest damage is region 1 or IDR due to deformations and damage caused by the impact of the jet on the material. In addition, this is the region where embedded particles were detected. On the other hand, the material at the bottom has lower roughness values in region 4 due to the protection of the material at the top.

On the other hand, it can be observed that regions 2 and 5, corresponding to SCR, do not have values lower than those recorded in zones 3 and 6 as RCR. This indicates the existence of two zones because the jet still has enough kinetic energy to make the cut without the appearance of striations.

### 3.2. Holes Evaluation

#### 3.2.1. Roundness Deviation

Figure 14 shows the data obtained from roundness deviations for each material and the total average of both materials. Readers can also find Appendix C with measured data and its standard deviation. In this way, the results can be analyzed separately.

Figure 14a shows the data for UNS A97075/CFRP configuration. The data show that in all tests the deviation is higher for CFRP, even though it is the material located at the bottom of the stack. This is due to the fact that the erosion and removal of composite materials is different from that produced in metallic materials. Thus, in CFRP the particles weaken and remove the matrix of the compound to subsequently break the fibers of the adjacent zone and in Al the process of material elimination takes place due to the micromachining produced by the edges of the abrasive particles, being more homogeneous the elimination of material in metallic materials. [2]. This phenomenon, combined with the material’s resistance to jet dispersion as a result of the loss of energy after cutting the aluminum, leads to an increase in the deflection in this material. This deflection increases considerably as WP decreases and TFR increases, which makes sense because these are tests with lower shear power.

On the other hand, Figure 14b shows the results of CFRP/UNS configuration A97075. In this particular case, the deviations follow a similar relationship to that of the previous case in terms of parameter influence, although it is true that the difference in the measured values is high. Thus, although in this case the aluminum is at the bottom of the pile, it seems that the expansion of the water jet does not deform the entry zone due to the differences in terms of removal of material explained in the previous paragraph. This results in homogeneous deviations in roundness between the two configurations, which favors a subsequent joint by means of rivets.

#### 3.2.2. Cylindricity Deviation

Cylindricity deviation was also measured with two measures for each material and configuration. However, due to the nature of the test, only one value results as output. Appendix D contains all collected data. Nevertheless, an ANOVA description of variables influence is shown in Figure 15.

The ANOVA analysis carried out shows that the parameters that have the greatest influence on the formation of the deviations are WP and TFR. Specifically, Figure 15 shows that the UNS configuration A97075/CFRP has lower cylindricity deviation values, which is in good agreement with the taper values obtained. This is due to the close relationship between both parameters. In order to offer a better correlation of results, the profiles measured for test 11 are presented as an example (Figure 16).

A more detailed description of the data reflects that cylindricity decreases as TFR decreases and WP increases. Specifically, Figure 15a reveals that TFR has a superior influence when the alloy is at the inlet of the material which reflects the importance of employing reduced feed rates to prevent its formation. As for Figure 16, both (a) and (b) show CFRP on the bottom and UNS A97075 on the top of the cone. It can be observed how it affects the energy loss to the generated hole, especially in Figure 16a.

#### 3.2.3. Straightness Deviation

In this case, there is no distinction between materials and straightness was evaluated throughout the entire profile. Thus, Figure 17 shows a comparison between the values obtained for the two configurations. However, Appendix E shows all measured data with the numerical value of the standard deviation.

As a general conclusion, a higher water jet drilling capacity means less straightness deviation. It is also observed that the CFRP/UNS A97075 configuration shows better results for the same test number except for the last three tests which, due to their lower drilling capacity due to the use of lower WP and TFR, are not able to maintain a uniform cutting profile of the aluminum alloy and therefore cannot maintain straightness along the hole.

The results reveal that the data in configuration UNS A97075/CFRP are slightly lower than those recorded in configuration CFRP/UNS A97075. In addition, it should be noted that for high pressures the straightness deviation increases when the compound is located at the top.

On the other hand, it should be noted that the standard deviation presented by the results is high, which makes it difficult to establish relationships between the results. This major standard deviation is directly related to the greater cylindricity deviation of some tests.

## 4. Conclusions

A study has been carried out on the influence of the parameters of the abrasive water jet on the quality of straight cuts and holes in composite materials and aeronautical aluminum. Based on this, the following conclusions can be drawn:
The machining of straight cuts has revealed that thermal damage is eliminated and the appearance of delamination in CFRP is reduced. Thus, for the selected parameters, no delamination was found in the mechanized test samples.The proposed kerf taper measurement method was validated for measurement in stacks. The results show the influence of the selected parameters obtaining the best results for high TFR and AMFR for both configurations, especially USN A97075/CFRP, with CFRP being the material with the highest kerf taper. On the other hand, the CFRP/UNS A97075 configuration has lower microgeometric deviations for the three evaluated parameters due to the lower loss of jet energy.Ra is in all cases below 7 μm, although this value is specific for tests 9, 10, 11 and 12. The functional holes show a lower roughness for both materials in any configuration. Nevertheless, it appears that the UNS A97075/CFRP configuration offers a better roughness of the holes.The study of surface quality has revealed that the IDR zone of the second material (region 4) is attenuated from impacts of particle and EAZ impacts. On the other hand, the presence of RCR was not detected.The measurements obtained of roundness present a greater deviation at the entrance of the drill due to the IDR zone in region 1, independently of the selected configuration, although it is true that the CFRP/UNS A97050 configuration presents values around 200% lower for the tests with lower penetration power (9, 10, 11 and 12).The influence of kerf taper on cylindricity deviations was reflected through the evaluated profiles, recording that the parameters with the greatest influence on its formation are WP and TFR. In this case, the configuration UNS A97075/CFRP presents better results of cylindricity.The straightness deviations did not allow consolidated conclusions to be drawn due to the high standard deviationHowever, it can be seen once again that tests 9, 10, 11 and 12 have higher values.


Finally, it should be noted that this process does not generate burrs in metallic materials due to its abrasive nature or thermal gradients that damage the material. On the other hand, it should be noted that each configuration has different characteristics, but it is the UNS A97075/CFRP configuration that presents the best results in terms of macro and microgeometric deviations.

## Figures and Tables

**Figure 1 materials-12-00107-f001:**
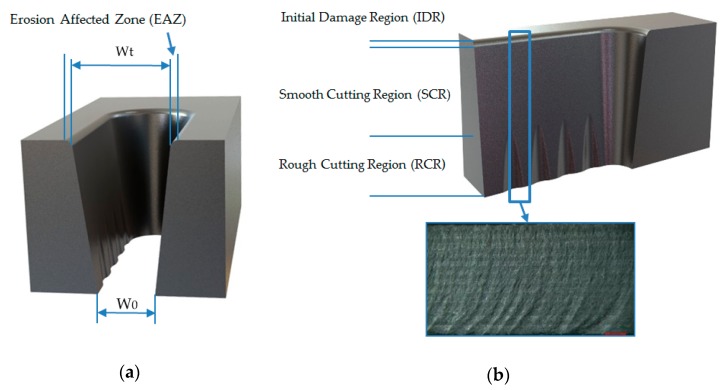
Scheme with the main defects associated with Abrasive Waterjet Machining (AWJM): (**a**) Erosion affected zone and kerf taper defined by inlet width (Wt) and outlet width (Wb); (**b**) different roughness zones that can be formed in AWJM.

**Figure 2 materials-12-00107-f002:**
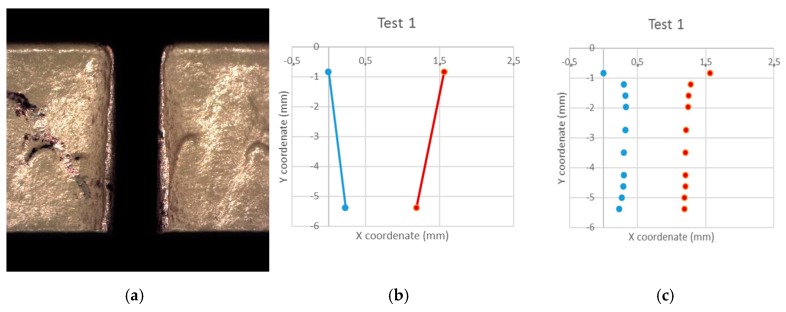
Proposal of measurement of the kerf taper from: (**a**) Stereoscopic Optical Microscopy (SOM) image; (**b**) geometry discretization; (**c**) traditional kerf assumption.

**Figure 3 materials-12-00107-f003:**
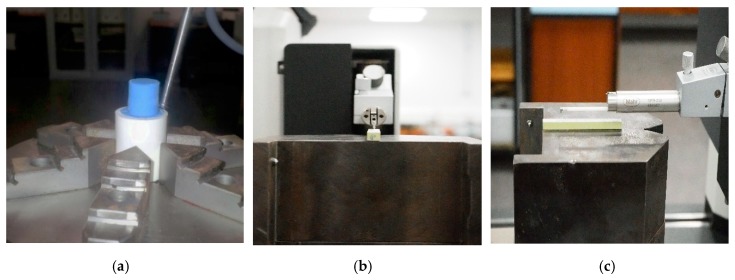
(**a**) Measure of replica to obtain geometrical results; (**b**) measure of roughness of UNS A97075, front view; (**c**) measure of roughness of UNS A97075, side view.

**Figure 4 materials-12-00107-f004:**
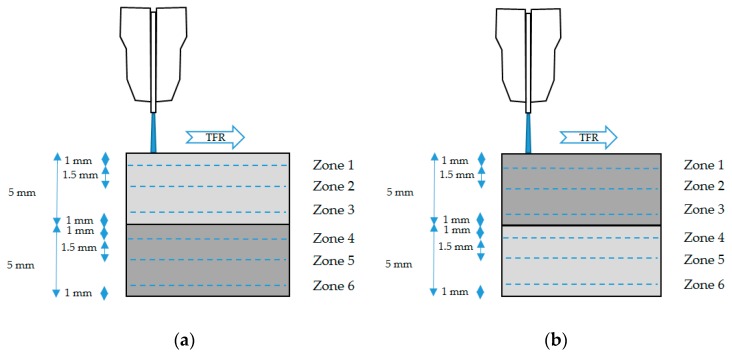
Schematic representing the roughness measurement zones and the distance between them for: (**a**) UNS A97075/CFRP configuration; (**b**) CFRP/UNS A97075 configuration.

**Figure 5 materials-12-00107-f005:**
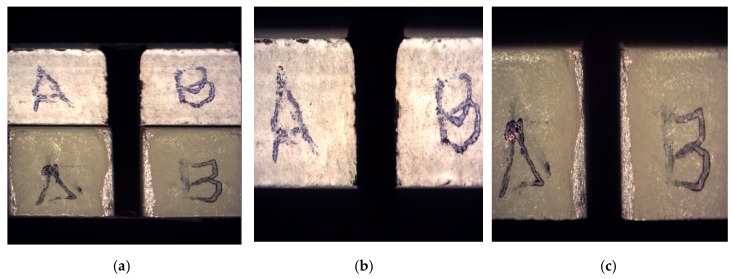
SOM image of the cutting front in: (**a**) Stack CFRP/UNS A97075; (**b**) CFRP plate; (**c**) UNS A97075 plate.

**Figure 6 materials-12-00107-f006:**
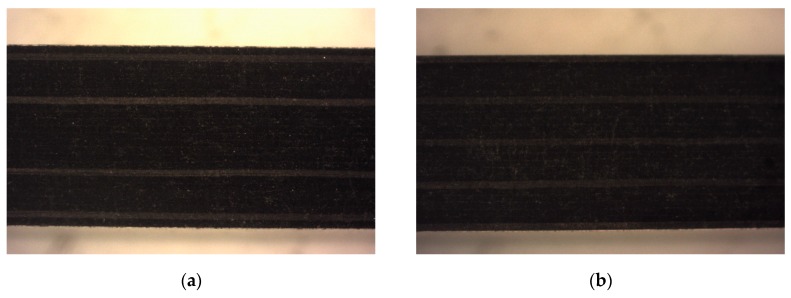
SOM image of CFRP profile. Test 2. Water pressure (WP) = 2500 bar, traverse feed rating (TFR) = 15 mm/min and abrasive mass flow rate (AMFR) = 340 gr/min for: (**a**) UNS A97075/CFRP; (**b**) CFRP/UNS A97075.

**Figure 7 materials-12-00107-f007:**
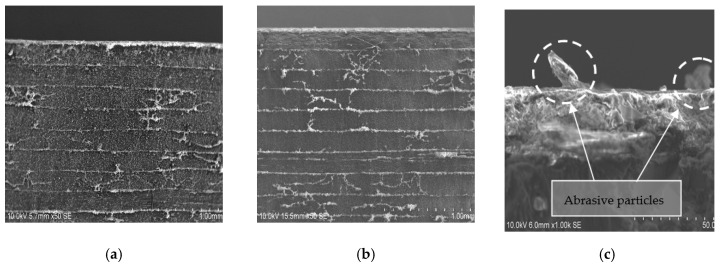
SEM image of CFRP profile. Test 2. WP = 2500 bar, TFR = 15 mm/min and AMFR = 340 gr/min for (**a**) UNS A97075/CFRP; (**b**) CFRP/UNS A97075; (**c**) Abrasive particles in the interface over CFRP.

**Figure 8 materials-12-00107-f008:**
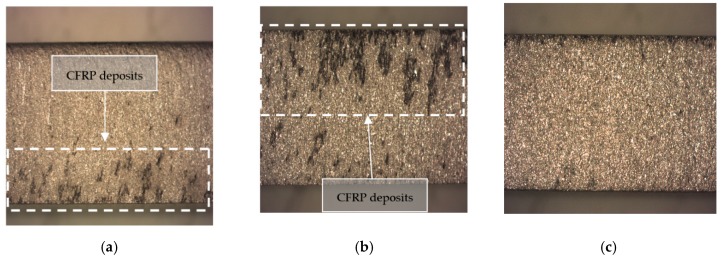
Profile SOM of UNS A97075 from: (**a**) UNS A97075/CFRP. Test 2. WP = 2500 bar, TFR = 15 mm/min and AMFR = 340 gr/min; (**b**) CFRP/UNS A97075. Test 2. WP = 2500 bar, TFR = 15 mm/min and AMFR = 340 gr/min; (**c**) CFRP/UNS A97075. Test 11. WP = 1200 bar, TFR = 45 mm/min and AMFR = 170 gr/min.

**Figure 9 materials-12-00107-f009:**
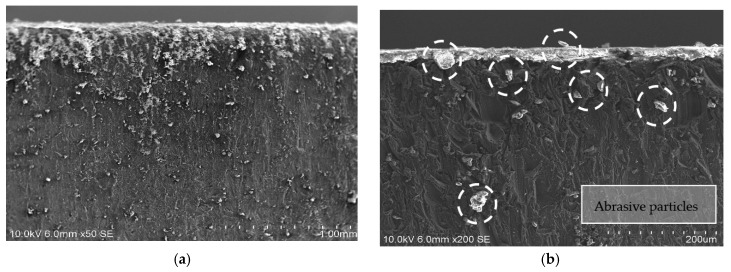
Test 2 Scanning Electron Microscopy (SEM) evaluation: (**a**) Abrasive imbued into UNS A97075; (**b**) abrasive particles in the interface and channel created over the material; (**c**) remains of carbon and point of EDS; (**d**) EDS results with a peak on the carbon.

**Figure 10 materials-12-00107-f010:**
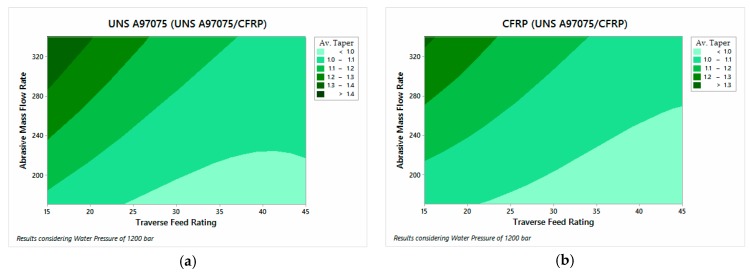
Average kerf taper for USN A97075/CFRP configuration: (**a**) UNS A97075; (**b**) CFRP.

**Figure 11 materials-12-00107-f011:**
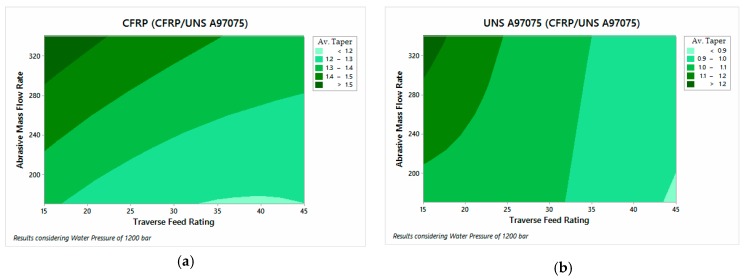
Average kerf taper for CFRP/USN A97075/ configuration: (**a**) CFRP; (**b**) UNS A97075.

**Figure 12 materials-12-00107-f012:**
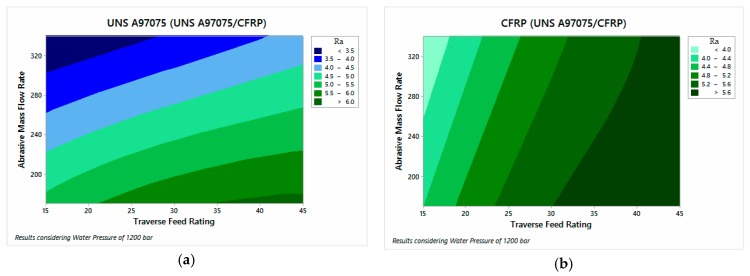
Roughness for UNS A97075/CFRP configuration: (**a**) UNS A97075 and (**b**) CFRP.

**Figure 13 materials-12-00107-f013:**
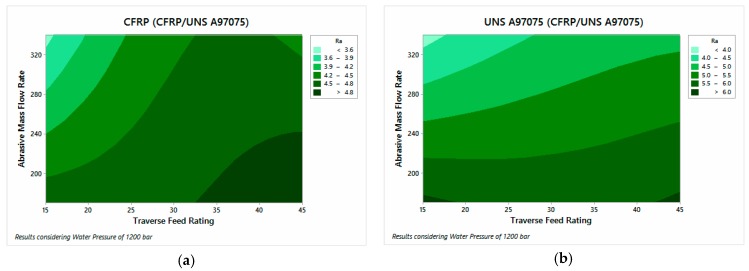
Roughness for CFRP/USN A97075 configuration: (**a**) CFRP and (**b**) UNS A97075.

**Figure 14 materials-12-00107-f014:**
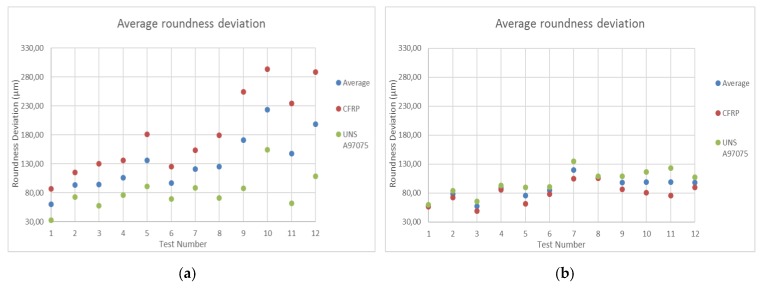
Average roundness: (**a**) UNS A97075/CFRP configuration; (**b**) CFRP/UNS A97075 configuration.

**Figure 15 materials-12-00107-f015:**
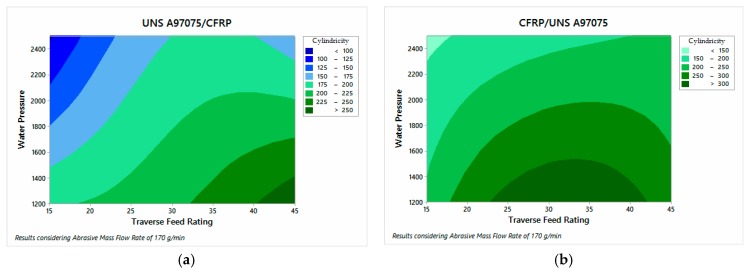
Cylindricity results on: (**a**) UNS A97075/CFRP configuration; (**b**) CFRP/UNS A97075 configuration.

**Figure 16 materials-12-00107-f016:**
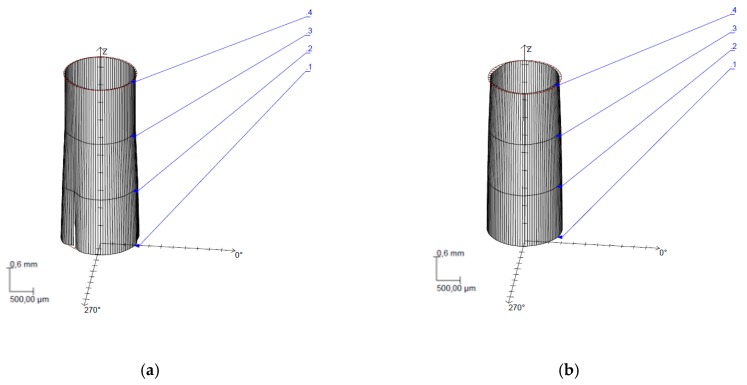
Cylindricity deviations. Test 11. WP = 1200 bar, TFR = 45 mm/min and AMFR = 170 gr/min for: (**a**) UNS A97075/CFRP configuration; (**b**) CFRP/UNS A97075 configuration.

**Figure 17 materials-12-00107-f017:**
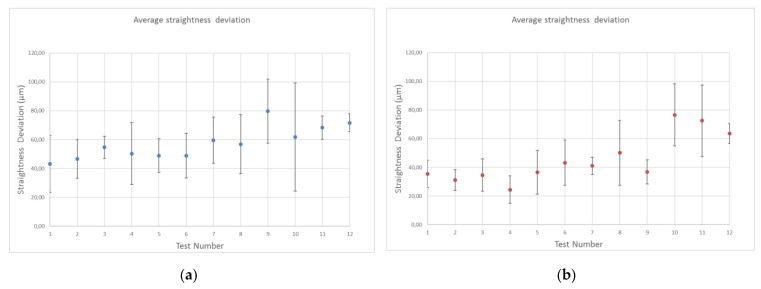
Straightness deviations results on: (**a**) UNS A97075/CFRP configuration; (**b**) CFRP/UNS A97075 configuration.

**Table 1 materials-12-00107-t001:** Comparative table with AWJM experimental studies on aeronautical materials.

Material	Thickness	Experimental	Parameters	Main Finding	Authors
CFRP/Ti-6Al-4V	10/11 mm	Straight Cuts	WP, TFR, AMFR	Taper analysis in stacks	Alberdi et al. [10]
CFRP	6 mm	Holes (6.35 mm)	WP, AMFR, SoD	Reduction of delamination	Phapale et al. [23]
CFRP	1.2 mm	Straight Cuts	WP, TFR, AMFR, SoD	Defect analysis	Schwartzentruber et al. [24]
CFRP	1.2 mm	Piercings	WP, TFR, AMFR, SoD	Piercing formation and delamination analysis	Schwartzentruber et al. [25]
Al 7075	7 mm	Straight Cuts	WP, TFR, SoD	Surface roughness analysis	Ahmed et al. [26]
GFRP	3.5 mm	Straight Cuts	WP, TFR, AMFR, SoD	Surface roughness analysis	Ming Ming et al. [27]
CFRP	6/12 mm	Straight Cuts	WP, AMFR	Taper and surface roughness analysis	Alberdi et al. [20]
CFRP	10.4 mm	Straight Cuts	WP, TFR, SoD	Taper analysis	El-Hofy et al. [16]
Ti-6Al-4V	5 mm	Straight Cuts	WP, TFR, AMFR	Taper and surface roughness analysis	Gnanavelbabu et al. [28]
GFRP	4 mm	Holes (10 mm)	WP, AMFR, SoD	Surface roughness and MRR	Prasad et al. [29]

**Table 2 materials-12-00107-t002:** Carbon Fiber Reinforced Plastic (CFRP) pieces features.

Type of Material	Composition	Production Method	Technical Specification
Layers of unidirectional carbon fiber with epoxy resin matrix and a symmetrical stacking sequence of (0/90/45/-45/45/-45)	Intermediate module fiber (66%) and epoxy resin (34%)	Pre-preg and autoclaved at 458° ± 5° at a pressure of 0.69 MPa	AIMS-05-01-002

**Table 3 materials-12-00107-t003:** Parameters used for each configuration.

Test	WP (bar)	TFR (mm/min)	AMFR (g/min)
1	2500	15	170
2	2500	15	340
3	2500	30	170
4	2500	30	340
5	2500	45	170
6	2500	45	340
7	1200	15	170
8	1200	15	340
9	1200	30	170
10	1200	30	340
11	1200	45	170
12	1200	45	340

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
