# Peer review of "Influence of Abrasive Waterjet Parameters on the Cutting and Drilling of CFRP/UNS A97075 and UNS A97075/CFRP Stacks"

_materials, 2018, doi:10.3390/ma12010107_

Reviewer 1 Report

This is a good paper.  I have to suggestions:

  All graphs should have units on the axis

 It would be useful to compare the results of the new method of estimating the proposed kerf taper measurement methods with results from the more traditional method.

Author Response

The explanation and comparison of the proposed method with the traditional method in the methodologies has been improved. In addition, references have been added and results improved.

Reviewer 2 Report

In the present work the authors investigated the influence of the abrasive waterjet working parameters on the cutting and drilling of hybrid stacks, made of UNS A97075 and carbon fibre reinforced polymer. In particular, several experimental tests were carried out, varying the process parameters, such as the final water pressure, the abrasive mass flow and the transverse feed capacity, and measuring the quality of the obtained surface in terms of kerf taper, particle lodging and surface roughness for the linear cut machining, or the roundness deviation, the cylindricity deviation and the straightness deviation for the holes drilling, in order to find a parameters set able to minimize the machining induced damage. The topic of the present work is surely interesting, since this kind of machining technology is more and more adopted in different industrial fields, but the article can be accepted for publication only after the following revisions are embedded in the text:

The English language is quite good, a slight review should be carried out to avoid typos and to rectify some terms that should be chosen more appropriately.

More information should be provided in the materials and methods section, such as the diameter of the nozzle; more details should also be given on the type of abrasive dispersed in the working fluid.

More details on the methodology adopted for the holes replication (page 5 line 146) should be reported.

Which pressure was considered in Fig. 9 -13?

The same scale should be assumed for the legends of the diagrams that concern the same measurements, for example those in Figs. 9 a, 9 b, 10 a and 10 b, in order to simplify the comparison of the results.

The authors implemented a new methodology to measure the kerf taper, but they should discuss in a deeper way the relevant validation, making a comparison with the results obtained through a well-established method.

The number of repetition and the obtained standard deviation should be reported for each kind of experimental test.

Some important contributions are lacking in the introduction section; for instance, the authors are invited to consider the works: “Sorrentino, L., Turchetta, S., Bellini, C. A new method to reduce delaminations during drilling of FRP laminates by feed rate control (2018) Composite Structures, 186, pp. 154-164. DOI: 10.1016/j.compstruct.2017.12.005” and “Saleem, M., Zitoune, R., El Sawi, I., Bougherara, H. Role of the surface quality on the mechanical behavior of CFRP bolted composite joints (2015) International Journal of Fatigue, 80, pp. 246-256. DOI: 10.1016/j.ijfatigue.2015.06.012”.

Author Response

-  More information should be provided in the materials and methods section, such as the diameter of the nozzle; more details should also be given on the type of abrasive dispersed in the working fluid

-  Aluminum information and properties and CFRP information have been added to Table 2. The rest of the fixed parameters (abrasive and part blast distance) appear in the methodology and the blast diameter in the results (Taper section).

-  More details on the methodology adopted for the holes replication (page 5 line 146) should be reported.

-  The explanation of the process was improved and an image of the process was added.

-   Which pressure was considered in Fig. 9 -13?. Poner en cada apartado las gráficas según una de las presiones, la más parecida.

- All graphs have been placed at 1200 bar pressure as it offers better results (the aim of the article is to try to reduce macro and microgeometric deviations). On the other hand, in cylindricity, the abrasive flow rate used has been indicated.

-  The same scale should be assumed for the legends of the diagrams that concern the same measurements, for example those in Figs. 9 a, 9 b, 10 a and 10 b, in order to simplify the comparison of the results.

- It has been tried to solve with the software, but could not be modified. In any case, the obtaining of data in methodology has been improved and the results have been improved.

- The authors implemented a new methodology to measure the kerf taper, but they should discuss in a deeper way the relevant validation, making a comparison with the results obtained through a well-established method. Modificar la misma gráfica para solo mostrar la distancia arriba y abajo, ponerla al lado para discutir que la información del centro del corte es importante.

- The explanation of how the new method was obtained has been improved and a reference defending the elimination of the IDR zone from the taper results has been included in the results.

- The number of repetition and the obtained standard deviation should be reported for each kind of experimental test.

- The number of repetitions (4 per test) has been written in the experimental methodology and the standard deviation of all the experimental ones has been added in appendices.

- Some important contributions are lacking in the introduction section; for instance, the authors are invited to consider the works: “Sorrentino, L., Turchetta, S., Bellini, C. A new method to reduce delaminations during drilling of FRP laminates by feed rate control (2018) Composite Structures, 186, pp. 154-164. DOI: 10.1016/j.compstruct.2017.12.005” and “Saleem, M., Zitoune, R., El Sawi, I., Bougherara, H. Role of the surface quality on the mechanical behavior of CFRP bolted composite joints (2015) International Journal of Fatigue, 80, pp. 246-256. DOI: 10.1016/j.ijfatigue.2015.06.012”.

- The main conclusion of Sorrentino is added where the AWJM process extends the fatigue life of CFRP rivets. On the other hand, Saleem is added as an author investigating that conventional drilling in CFRP damages the matrix and results in a worse surface finish than AWJM.

Reviewer 3 Report

The paper investigates the influence of abrasive waterjet parameters through hole drilling operation of a hybrid composite metal materials cfrp/uns a97075 and uns a97075/cfrp stacks. The metal part is an aluminium material. The authors studied the influence of abrasive mass flow rate, traverse feed rate and water pressure. The authors evaluated several machining outputs such as kerf taper, surface quality in different cutting regions, hole roundness, cylindricity and straightness. the authors claim that AWJM minimize deviations, defects, and damage in the drilled holes.

The authors have conducted a good literature review however, it does not highlight any in depth findings of previous studies. The authors should create a table that summarises previous studies (at least 10 studies) on AWJ drilling of metals, composites and composite metal stack materials, showing the type of material used, the cutting parameters used, the studied parameters and the findings as shown below:

Material type, thickness and hole size drilled

Cutting parameters

Studied parameters

Main findings

References

Does the authors have any data regarding the mechanical properties of the two materials, if so could you please include a table showing their mechanical properties for the sake of comparison and to give the readers a better idea on each material used in the study.

The authors then should make conclusions or report findings from those studies in the literature to support their case.

Why did the authors choose 8 mm hole size for their tests, please support with references or is it recommended by some company/industry?

163-167 rephrase and make it shorter, it is repetitive.

170- remove the first sentence as it is not adding any value to the manuscript. Start from Figure 6…

Figure 6, please highlight with arrows what are you trying to show to the readers. It is not clear what are you trying to show us without having any details on the images.

184-1865: please support you claim with references

Figure 7 and 8 same as figure 6, please show details of what are you trying to tell us from the images.

Please explain how you measured surface roughness, show details, images if possible.

I am a bit surprised that the roughness of Al is close to that of CFRP, usually Al roughness is much lower.  please compare your results of roughness with results from previous studies stack or individual Al and CFRP materials.

266, the width of the cut not the wide of cut

267: is this your own speculation or it was observed in previous studies too? Please support with references if possible.

345: any clue why the deviations are very high among measured data? Could there be other influencing parameters that were not analysed in the current study which may have affected the results.

Please describe how you measured each parameter in the materials and method sections

For roundness, please describe the way the material is removed/cut from composite and Al to explain the difference mechanism which each material undergo due to AWM.

Author Response

- The authors have conducted a good literature review however, it does not highlight any in depth findings of previous studies. The authors should create a table that summarises previous studies (at least 10 studies) on AWJ drilling of metals, composites and composite metal stack materials, showing the type of material used, the cutting parameters used, the studied parameters and the findings as shown below:

Material type, thickness and hole size drilled

Cutting parameters

Studied parameters

Main findings

References 

-  Table 1 has been added with the comparison.

-   Does the authors have any data regarding the mechanical properties of the two materials, if so could you please include a table showing their mechanical properties for the sake of comparison and to give the readers a better idea on each material used in the study. (igual que antes, poner la table)

-   Aluminum information and properties and CFRP information have been added to Table 2.

- The authors then should make conclusions or report findings from those studies in the literature to support their case.

-   Added with table 1, where experimental studies are shown.

- Why did the authors choose 8 mm hole size for their tests, please support with references or is it recommended by some company/industry?

-   Added in methodology the justification.

-  163-167 rephrase and make it shorter, it is repetitive.

-   The text has been modified.

-  170- remove the first sentence as it is not adding any value to the manuscript. Start from Figure 6…

-  The text has been modified.

- Figure 6, please highlight with arrows what are you trying to show to the readers. It is not clear what are you trying to show us without having any details on the images.

- This is an image that shows the absence of delamination so you can not point anything out on the image. On the other hand, the abrasive has been identified.

- 184-185: please support you claim with references

- The explanation has been extended and a reference has been added to support the idea.

- Figure 7 and 8 same as figure 6, please show details of what are you trying to tell us from the images.

- It has been indicated in the images.

- Please explain how you measured surface roughness, show details, images if possible.

added in methodology.

-  I am a bit surprised that the roughness of Al is close to that of CFRP, usually Al roughness is much lower.  please compare your results of roughness with results from previous studies stack or individual Al and CFRP materials. Pedro

- As shown in Table 1, no AWJM drilling studies with two roughness materials have been found. Therefore, results could not be compared.  Regarding drilling studies for a single material, most CFRP studies are not divided by zones (IDR, SCR and SCR), although a minimum Ra of about 7 micrometres are obtained (Alberdi et al), which is related to the study carried out.

- 266, the width of the cut not the wide of cut.

- Modified.

- 267: is this your own speculation or it was observed in previous studies too? Please support with references if possible.

- The discussion of results has been clarified in the contour graph and an attempt has been made to improve the discussion of results.

- 345: any clue why the deviations are very high among measured data? Could there be other influencing parameters that were not analysed in the current study which may have affected the results.

- The high dispersion is due to the uncut triangle that is produced at the exit of the drill by the nature of the process along with the low kinetic energy. This defect is shown in figure 16 a.

- Please describe how you measured each parameter in the materials and method sections

- Added in methodology.

- For roundness, please describe the way the material is removed/cut from composite and Al to explain the difference mechanism which each material undergo due to AWM.

- Modified and added

Round  2

Reviewer 3 Report

All queries have been answered, the manuscript can now be accepted